# Harmine and Piperlongumine Revert TRIB2-Mediated Drug Resistance

**DOI:** 10.3390/cancers12123689

**Published:** 2020-12-09

**Authors:** Susana Machado, Andreia Silva, Ana Luísa De Sousa-Coelho, Isabel Duarte, Inês Grenho, Bruno Santos, Victor Mayoral-Varo, Diego Megias, Fátima Sánchez-Cabo, Ana Dopazo, Bibiana I. Ferreira, Wolfgang Link

**Affiliations:** 1Centre for Biomedical Research (CBMR), Universidade do Algarve, Campus of Gambelas, Building 8, Room 1.12, 8005-139 Faro, Portugal; scmachado@ualg.pt (S.M.); afl.silva@campus.fct.unl.pt (A.S.); alcoelho@ualg.pt (A.L.D.S.-C.); giduarte@ualg.pt (I.D.); a63461@ualg.pt (I.G.); a66853@ualg.pt (B.S.); 2Algarve Biomedical Center (ABC), Universidade do Algarve, Campus de Gambelas, 8005-139 Faro, Portugal; 3Regenerative Medicine Program, Department of Biomedical Sciences and Medicine, Universidade do Algarve, Campus de Gambelas, 8005-139 Faro, Portugal; 4Instituto de Investigaciones Biomédicas “Alberto Sols” (CSIC-UAM), Arturo Duperier 4, 28029 Madrid, Spain; vmayoral@iib.uam.es; 5Confocal Microscopy Unit, Biotechnology Program, Spanish National Cancer Research Centre (CNIO), 28029 Madrid, Spain; dmegias@cnio.es; 6Vascular Pathophysiology Area, Centro Nacional de Investigaciones Cardiovasculares (CNIC), 28029 Madrid, Spain; fscabo@cnic.es (F.S.-C.); adopazo@cnic.es (A.D.)

**Keywords:** TRIB2, cancer, drug resistance, FOXO, BEZ235, harmine, piperlongumine

## Abstract

**Simple Summary:**

Poor survival and treatment failure of patients with cancer are mainly due to resistance to therapy. Tribbles homologue 2 (TRIB2) has recently been identified as a protein that promotes resistance to several anti-cancer drugs. In this study, RNA sequencing and bioinformatics analysis were used with the aim of characterizing the impact of TRIB2 on the expression of genes and developing pharmacological strategies to revert these TRIB2-mediated changes, thereby overcoming therapy resistance. We show that two naturally occurring alkaloids, harmine and piperlongumine, inverse the gene expression profile produced by TRIB2 and sensitize cancer cells to anti-cancer drugs. Our data suggest that harmine and piperlongumine or similar compounds might have the potential to overcome TRIB2-mediated therapy resistance in cancer patients.

**Abstract:**

Therapy resistance is responsible for most relapses in patients with cancer and is the major challenge to improving the clinical outcome. The pseudokinase Tribbles homologue 2 (TRIB2) has been characterized as an important driver of resistance to several anti-cancer drugs, including the dual ATP-competitive PI3K and mTOR inhibitor dactolisib (BEZ235). TRIB2 promotes AKT activity, leading to the inactivation of FOXO transcription factors, which are known to mediate the cell response to antitumor drugs. To characterize the downstream events of TRIB2 activity, we analyzed the gene expression profiles of isogenic cell lines with different TRIB2 statuses by RNA sequencing. Using a connectivity map-based computational approach, we identified drug-induced gene-expression profiles that invert the TRIB2-associated expression profile. In particular, the natural alkaloids harmine and piperlongumine not only produced inverse gene expression profiles but also synergistically increased BEZ235-induced cell toxicity. Importantly, both agents promote FOXO nuclear translocation without interfering with the nuclear export machinery and induce the transcription of FOXO target genes. Our results highlight the great potential of this approach for drug repurposing and suggest that harmine and piperlongumine or similar compounds might be useful in the clinic to overcome TRIB2-mediated therapy resistance in cancer patients.

## 1. Introduction

Many cancer patients fail to respond to anti-cancer treatment due to intrinsic or acquired resistance, widely described for various types of cancer [1]. Resistance to anti-cancer therapy is the major obstacle to improving the clinical outcome for cancer patients and to reap the full benefits of targeted therapies and immunotherapies. Therefore, identification of the molecular mechanisms involved in drug resistance is of great clinical and economic importance. We have previously characterized a novel molecular mechanism of therapy resistance in melanoma mediated by the kinase-like protein Tribbles homologue 2 (TRIB2) [2]. TRIB2 encodes one of three members of the human Tribbles family proteins that are highly conserved throughout evolution [3,4]. The three human Tribbles (TRIB) pseudokinases are homologues of the Drosophila melanogaster pseudokinase termed Tribbles, which controls ovarian border cell and neuronal stem cell physiology [5,6]. The human TRIB1, -2 and -3 orthologues contain a catalytically impaired pseudokinase domain, and instead of directly phosphorylating target proteins, they act as adaptors in signaling pathways for important cellular processes. Tribbles proteins are unique in combining a kinase domain with an adjacent C-terminal motif (COP-1) that engages the ubiquitin E3 ligase machinery [7]. We identified TRIB2 in a genetic screening aimed at the identification of repressor proteins of FOXO transcription factors [8], the major downstream transcriptional effectors of the PI3K/AKT pathway [9]. Furthermore, our studies identified TRIB2 as a novel oncogene and as a biomarker in melanoma [8], whose expression correlates with disease stage [10].

TRIB2 has been established as an oncogene in several other solid tumors, such as lung and liver cancer [5,11,12]. We and others showed that TRIB2 confers resistance to several anti-cancer drugs [13,14,15,16], including PI3K/mTOR inhibitors [2], gemcitabine, dacarbazine and temozolomide [17]. TRIB2 preferentially binds to catalytically inactive, non-threonine 308 phosphorylated AKT1 in vitro and increases endogenous AKT phosphorylation at the hydrophobic motif (serine 473) in human cells [18]. TRIB2-mediated AKT activation results in FOXO inhibition, which, in turn, reduced expression of FOXO targets, including genes involved in apoptosis. Consequently, drug-induced apoptosis is attenuated by TRIB2. Among the drugs whose efficiency was found to be affected by the level of TRIB2 expression was the imidazoquinoline derivative BEZ235. This ATP-competitive inhibitor of PI3K and mTOR kinases has undergone numerous clinical trials [19]. Several other PI3K and mTOR inhibitors have been approved for clinical use against different cancer types [20]. Interfering with TRIB2 activity might be a therapeutic strategy for the treatment of diverse tumor types and, particularly, to overcome therapy resistance. By analyzing TRIB2-induced and inverse transcriptional signatures, we expect to further characterize TRIB2 action at the molecular level and discover pharmacological ways to interfere with its activity. Here, we use RNA sequencing-based transcriptional profiling of isogenic cells lines with different TRIB2 statuses in the presence or absence of PI3K/mTOR inhibition to identify sets of differentially expressed genes (DEGs) and connectivity map-based algorithms to identify drugs or chemical compounds capable of inducing reverse transcriptional signatures.

Our results show that several small-molecule compounds, including harmine (HAR) and piperlongumine (PIP), can affect TRIB2-mediated transcriptional signatures and, in line with this observation, can induce the nuclear translocation of FOXO3 and its consequent activation. The identified compounds or similar molecules might have the potential to attenuate TRIB2-mediated therapy resistance in several human cancers.

## 2. Results

### 2.1. TRIB2-Induced Transcriptional Signature

We previously reported TRIB2-mediated resistance to several anti-cancer agents, including the dual PI3K/mTOR inhibitor BEZ235 [2]. To gain insight into the molecular mechanism by which TRIB2 mediates resistance, we analyzed global transcriptional changes using next-generation sequencing of total RNA (RNA-seq) in U2OS osteosarcoma cells, a cellular model which we have previously used to analyze the function of TRIB2 [2,8]. The expression of TRIB2 in U2OS is very low at the RNA level and is undetectable at the protein level, as measured by Western blot analysis (Appendix A). The expression level of TRIB2 considerably varies among cell (Appendix A) and tumor types, with melanoma and glioblastoma being the cancers that most strongly express TRIB2 (Appendix A). TRIB2 overexpression in the isogenic U2OS cell line (Appendix A) is well within the range of the physiological level of TRIB2 expression. By comparing U2OS TRIB2-overexpressing cells with mock-transfected cells in basal conditions and upon BEZ235 treatment, we obtained four lists of differentially expressed genes (DEGs) (Figure 1A), in which DEG5 and DEG6 are a subset of DEG1 and DEG2, respectively. Hierarchical clustering of the DEGs revealed complete segregation of untreated cells and cells treated with the dual PI3K/mTOR inhibitor BEZ235, independently of TRIB2 status (Figure 1B). TRIB2 overexpression resulted in upregulation and downregulation (log2 fold change 0.5 and false discovery rate (FDR) < 0.05) of 111 and 164 genes, respectively, constituting a total of 275 genes (DEG2) (Figure 1C and Appendix A). Treatment with BEZ235 resulted in changes in the expression levels of 2394 genes—1365 and 1029 upregulated and downregulated genes, respectively (DEG1) (Figure 1C and Appendix A). Moreover, TRIB2 overexpression in cells treated with BEZ235 resulted in differential expression of 2630 genes (Appendix A), from which 2068 were in common to BEZ235 treatment—1214 upregulated genes and 852 downregulated genes (DEG3)—and 113 genes were in common to TRIB2 overexpression, with 79 upregulated and 132 downregulated genes (DEG4). As expected, and validating our approach, TRIB2 was identified as the most upregulated gene in the experimental isogenic cell line with different TRIB2 statuses. This was verified by quantitative real-time PCR (RT-qPCR) and Western blotting analysis (Figure 1H and Appendix A). Besides, TRIB2-mediated downregulation of CCAAT/enhancer-binding protein α (CEBPA, Figure 1D) is consistent with previous studies [21,22] and further confirms the robustness of our approach. 

We also validated several of the top up- and downregulated genes mediated by TRIB2 overexpression in both untreated and BEZ235-treated cells (Figure 1D–G) by using RT-qPCR. The top genes were chosen from the list of differentially expressed genes (Figure 1C), from which we selected the 10 most upregulated and 10 most downregulated genes by TRIB2, as well as genes described to be involved in cancer proliferation and survival, namely Keratin 14 (KRT14), SRY-Box Transcription Factor 2 (SOX2), Formin-like protein 2 (FMNL2), BCL2-associated X, apoptosis regulator (BAX), TNF superfamily member 10 (TNFSF10), CEBPA, baculoviral IAP repeat-containing protein 7 (BIRC7), KiSS-1 metastasis suppressor (KISS1), Sestrin 2 (SESN2), MYCN Proto-Oncogene, BHLH Transcription Factor (MYCN) and BCL2 Binding Component 3 (PUMA) (Figure 1D–G) [23,24,25,26,27,28,29,30,31,32,33]. KRT14, SOX2 and FMNL2 were upregulated by TRIB2 and downregulated by BEZ235 treatment, while BAX, TNFSF10 and CEBPA were downregulated by TRIB2 (Figure 1D). Similarly, TRIB2 overexpression correlated with downregulation of BIRC7, while BEZ235 treatment upregulates its expression. In addition, we found that KISS1 is upregulated after BEZ235 treatment (Figure 1F–G). Our data also show that BEZ235 treatment downregulates SESN2 and MYCN (Figure 1F). 

Analysis using qRT-PCR revealed that the upregulation of MYCN, SOX2, Keratin 14 (KRT14) and Sestrin 2 (SESN2) in U2OS-TRIB2 cells is statistically significant (*p* < 0.05) compared to mock-transfected control (MOCK) cells (Figure 1H). On the other hand, PUMA and KISS1 gene regulation depends on TRIB2 status, being downregulated when TRIB2 was overexpressed (Figure 1I). Moreover, PUMA, KISS1 and FMNL2 transcriptional levels are modulated by BEZ235 (Figure 1I). Interestingly, TRIB2 overexpression appears to blunt BEZ235’s effect on these cells (Figure 1I), suggesting that TRIB2 counteracts PI3K/mTOR pathway inhibition.

Principal component analysis (PCA) of the logarithmized counts per million (log CPM) of the RNA-seq data showed that TRIB2 overexpression and BEZ235 treatment have a strong impact on gene expression (Appendix A). To evaluate the impact of TRIB2 and BEZ235 on the differential expression of the analyzed genes, we generated MA plots [34] to visualize the distribution of the DEGs (red dots) in terms of read counts (log CPM) and fold change (logFC) among the gene population (Appendix A). TRIB2 overexpression in untreated (Appendix A) and BEZ235-treated cells (Appendix A) significantly affected the expression of several genes (red dots). On the other hand, BEZ235 treatment (Appendix A) affected a greater number of genes compared to TRIB2 overexpression (Appendix A). The majority of the logFC magnitudes in both TRIB2-overexpressed and BEZ235-treated cells lie under two, which means that most of the expression levels of DEGs increased or decreased up to four times.

From the 2396 genes differentially regulated by BEZ treatment (DEG1) in cells with low TRIB2, 321 are BEZ-exclusive (Figure 1C) (DEG5). Enrichment analysis of the DEG5 gene list for kinase perturbations from the GEO database (obtained from the Enrichr tool [35,36]) resulted in the enrichment of the perturbation of various kinases, including inhibition of PI3Kα mutant, advanced glycosylation end-product-specific receptor (AGER), also known as RAGE, spleen tyrosine kinase (SYK), epidermal growth factor receptor (EGFR), ABL Proto-Oncogene 1, non-receptor tyrosine kinase (ABL1), cyclin-dependent kinase 19 (CDK19) and ataxia telangiectasia mutated (ATM) (Appendix A).

From the 275 genes differentially regulated by TRIB2 overexpression (DEG2) in untreated cells (Figure 1C), 155 are exclusively regulated by TRIB2 (DEG6). Enrichment analysis of the DEG6 gene list for kinase perturbations from the GEO database (obtained from the Enrichr tool) resulted in the enrichment of the perturbation of several kinases, including activation of Phospholipid Dependent Kinase 1 (PDK1), Glycogen Synthase Kinase 3 Alpha (GSK3α and IL2 Inducible T Cell Kinase (ITK) (Appendix A).

To understand the relevance of these genes in the context of Kyoto Encyclopedia of Genes and Genomes (KEGG) pathways, we performed an enrichment analysis on TRIB2- and BEZ235-induced transcriptional signatures with the Enrichr tool. Enrichment analysis of the list of TRIB2 upregulated transcripts showed a significant enrichment for genes involved in pathways in cancer and TNF signaling (Appendix A), alcoholism (Appendix A) and transcriptional misregulation in cancer (Appendix A). BEZ235 treatment upregulated transcripts and showed a significant enrichment for genes involved in pathways in cancer, TNF signaling and small-cell lung cancer (Appendix A).

Taken together, these data indicate that TRIB2 signature most significantly affects genes involved in cell survival and proliferation.

### 2.2. HAR and PIP Reverse TRIB2-Induced Expression Profiles

We hypothesize that TRIB2-mediated therapy resistance may be abolished or attenuated by pharmacological treatment capable of reversing TRIB2-mediated transcriptional signatures. Hence, drugs that inverse TRIB2 signatures are expected to resensitize cells to the effect of BEZ235 treatment on cell viability and proliferation. 

We took advantage of the TRIB2-induced gene expression profile (DEG2) to match it with connectivity map (cMAP) gene expression data and selected bioactive chemical compounds and US food and drug administration (FDA)-approved drugs that produce an inverse gene expression profile using pattern-matching algorithms of cMAP [37]. cMAP is a gene expression database obtained from experiments on cancer cell lines treated with approximately 5000 small-molecule compounds [37]. The list of candidate drugs was obtained by ruling out enrichment scores with P-values greater than 0.05. The enrichment scores are calculated based on the Kolmogorov–Smirnov statistic, a non-parametric rank statistic, also known as gene set enrichment analysis (GSEA), to interpret gene expression data [38,39]. The compounds harmine (HAR), piperlongumine (PIP), irinotecan and LM-1685 produced an inverse gene expression profile compared to the profile induced by TRIB2 (Figure 2A). Conversely, three compounds with anti-inflammatory properties, maprotiline, cromoglicic acid and crachidonyl trifluoromethane, produced a similar gene expression profile compared to the profile induced by TRIB2 overexpression (Appendix A). 

HAR is a β-carboline alkaloid isolated from the seeds of *Peganum harmala* and inhibits monoamine oxidase [40]. PIP is a natural amide alkaloid [41,42] isolated from the *Piper longum* Linn plant and is known to disrupt redox homeostasis by inhibiting glutathione S-transferase π (GSTπ) and carbonyl reductase 1 (CBR1) [42]. We validated the most differentially regulated genes upon HAR and PIP treatment retrieved in the cMAP database by qRT–PCR. Analysis of U2OS-TRIB2 cells treated with HAR showed that the genes LMCD1 and PCT1A are significantly downregulated (Figure 2B). Furthermore, PIP significantly downregulated the expression of tumor necrosis factor 6 (TNFAIP6) and GATA Binding Protein 3 (GATA3) (Figure 2C). On the contrary, ZNF3, together with the ANO2 and GNRH1 genes, is significantly upregulated after HAR and PIP treatment, respectively (Figure 2B,C). 

To test the hypothesis that HAR and PIP sensitize cells that are resistant to anti-cancer drugs, we first sought to determine the most effective concentrations for combined treatment with the PI3K/mTOR inhibitor BEZ235 and HAR or PIP. We used an MTT assay, which measures metabolic activity as an indicator of cell viability of U2OS-TRIB2 cells treated with serial two-fold dilutions of either BEZ235, HAR or PIP for 48 h (Appendix A) and 72 h (Appendix A). The dose-response plots at 72 h show growth inhibition by 50% as compared to DMSO (GI50), as measured using the MTT assay, produced by treatment with HAR, PIP and BEZ235 was achieved at 12.89 µM, 2 µM and 95.37 nM, respectively. To determine whether co-treatment of BEZ235 with HAR or PIP could synergistically affect BEZ235-treated cells, we inferred cell viability by MTT assay in U2OS-TRIB2 cells. Co-treatment of BEZ235 and HAR significantly decreased cellular metabolic activity at all concentrations, and combined ratios at 125 nM BEZ235/11.4 µM HAR and 250 nM BEZ235/22.8 µM HAR showed synergy, as depicted by a combination index below one (Figure 2D). Co-treatment of BEZ235 and PIP also significantly decreased cell viability, with synergy detected at 250 nM BEZ235/2.5 µM PIP and 500 nM BEZ235/5 µM PIP (Figure 2D). These results suggest that HAR and PIP together with BEZ235 synergistically increase the cytostatic/cytotoxic effects of BEZ235. To evaluate if the decrease in cellular metabolic activity at the synergic concentrations, determined by the MTT assays, was in part due to cell death, we performed a cell-death assay by trypan blue on the U2OS-TRIB2 cells treated with 250 nM BEZ235 and 22.8 µM HAR (the chosen synergic concentration) or 250 nM BEZ235 and 2.5 µM PIP (the chosen synergic concentration) for 72 h. In line with the MTT results, we found that combined drug treatment induces an increase in cell death compared to DMSO control (Figure 2E). Nevertheless, it also suggests that these drugs may also act by preventing cell proliferation (cytostasis) (Appendix A), as cell death did not increase as much as cellular metabolic activity decreased. Indeed, BEZ235 has been reported to exhibit more cytostatic effects than cytotoxicity effects on other cell lines [43]. Similarly, HAR exhibits cytostatic effects on cells [44], while PIP is known to have a strong cytotoxic effect [45].

To evaluate if the synergistic effect of HAR and BEZ235 or PIP and BEZ235 would affect TRIB2 modulation of its target genes, we performed RT-qPCR on cells with low and high levels of TRIB2 treated with BEZ235 alone (250 nM) or combined with HAR (22.8 µM) or PIP (2.5 µM) and evaluated the expression levels of genes regulated by both TRIB2 and HAR or PIP. TRIB2 overexpression prevented BEZ235 from significantly downregulating LMCD1, but when cells were treated with BEZ235 and HAR, LMCD1 was further downregulated, regardless of TRIB2 levels (Figure 2F). Our results indicate that HAR and PIP may help cells regain sensitivity to BEZ235 toxicity in the presence of high levels of TRIB2.

### 2.3. HAR and PIP Promote FOXO Nuclear Translocation

We previously identified TRIB2 as a FOXO suppressor protein promoting cytoplasmatic localization of FOXO3, leading to its inactivation [8]. To test if HAR and PIP would exert an opposite effect on FOXO, we treated a previously established reporter cell line (U2foxRELOC) with HAR or PIP. U2foxRELOC stably expresses a fluorescently labelled FOXO3 fusion protein and enables an image-based approach to monitor the subcellular localization of FOXO3 [46]. The treatment of the cells with the nuclear export inhibitor Leptomycin B (LMB) shifted the reporter protein almost entirely into the cell nucleus. Treatment of these cells with HAR and PIP for 1 h is sufficient to promote FOXO nuclear localization, similar to the positive control LMB (Figure 3A). We were able to quantify these results automatically by quantifying the green fluorescence within the area stained with 4′,6-diamidino-2-phenylindole (DAPI), which defined the cell nucleus and the extended cytoplasm. As shown in Figure 3B, both compounds scored 20% more positive cells than the negative control (DMSO). Next, we determined if the nuclear translocation of FOXO3 was due to the inhibition of the nuclear export receptor CRM-1, known to recognize the nuclear export sequence (NES) in the FOXO proteins. We treated U2nesRELOC cells [47] with 16 µM of HAR or 13 µM of PIP for 1 h and analyzed the subcellular distribution of the fluorescent signal. The results obtained from this experiment showed that treatment with HAR or PIP does not affect nuclear export through CRM-1 (Appendix A), suggesting that these agents interfere with the regulatory network upstream of FOXO3.

Since FOXO nuclear translocation is often accompanied by loss of phosphorylation on Ser253, a target residue of the upstream kinase AKT [48], we monitored the phosphorylation status of FOXO and AKT upon HAR and PIP treatment. As shown in Figure 3C, the exposure of cells to HAR for 1 h already affected the phosphorylation of AKT and FOXO3 but, at the same time, decreased the total amount of FOXO. Longer treatment periods provided a similar pattern of AKT and FOXO3 regulation (Figure 3C and Appendix A). These data show that HAR treatment may decrease FOXO protein stability and suggest that FOXO phosphorylation by AKT might not be the predominant mechanism of HAR-induced translocation of FOXO. Whereas PIP treatment for 1 h strongly induced the phosphorylation of AKT, longer exposure to PIP reverted the effect and led to decreased levels of AKT phosphorylation after 12 h (Figure 3D and Appendix A). Although PIP treatment also decreased the amount of FOXO3 protein, the net effect on FOXO phosphorylation is evident, suggesting that PIP impairs FOXO phosphorylation, and this effect seems to be regulated by the activation status of AKT.

Next, we investigated if the nuclear localization of FOXO induced by HAR or PIP would affect the transcription of FOXO target genes. We treated U2foxRELOC cells with HAR or PIP and performed RT-qPCR analysis using specific primers for known FOXO target genes. HAR and PIP treatment upregulated p21 and PUMA (Figure 3E), known to be regulated by FOXO3 [49,50]. However, as p21 and PUMA might be regulated by other transcription factors, including p53, we cannot exclude that additional mechanisms can contribute to or even represent the predominant mode of action of HAR and PIP. 

### 2.4. Validation Using CRISPR/Cas9 TRIB2 KO Cells

In order to analyze and validate the obtained data using a different cell line, we generated an alternative isogenic cell system in which we completely abolished the expression of TRIB2. We first extracted data from the GEO profiles [51] (Appendix A) and screened a panel of cell lines for TRIB2 mRNA levels and identified human malignant melanoma cells UACC-62 as the cell line with the highest level of endogenous of TRIB2. Next, we used the CRISPR/Cas9 system to genetically disrupt the TRIB2 locus in UACC-62, effectively generating TRIB2 knockout (KO) cells, as confirmed by the absence of TRIB2 protein in Western blot analysis (Appendix A). We performed an MTT assay (as described previously) and calculated GI50 values on parental UACC-62 cells following treatment with BEZ, HAR and PIP for 72 h. We obtained values of 20.4 nM, 9.06 uM and 2.14 uM for BEZ235, HAR and PIP, respectively (Appendix A). Following this, we assessed cell death by a trypan blue exclusion assay using the GI50 values obtained previously. These results show that co-treatment with BEZ235 and HAR (or PIP) tends to increase cell death, although this is not statistically significant, which suggests that the dominant effect of these drugs on the cells may be through arresting cell proliferation (cytostasis) and not solely cell death (cytotoxicity) (Appendix A). Next, we validated the previously observed synergy effect between these drugs in the U2OS cell line. We obtained a CI value of 0.5 using the 78.1 nM BEZ/25 µM HAR treatment combination in the UACC-62 cell line (Appendix A), which indicates a synergistic effect between both drugs (Appendix A). Moreover, we observed a significant reduction in cell viability upon combined treatment of HAR and BEZ at combinations of 39.1 nM BEZ/12.5 uM HAR and 19.5 nM BEZ/6.25 µM HAR, when compared to each treatment alone (Appendix A). Combined treatment of 39.1 nM BEZ and 3.13 uM PIP also showed a statistically significant decrease in cell viability when compared with single-drug treatments (Appendix A). Taken together, these data indicate that these drugs display cytotoxicity, but their dominant effect may be cytostatic, similar to the results obtained from the experiments in U2OS-TRIB2 cells (Figure 2D and Appendix A). In addition, we transfected isogenic UACC-62 cell lines with a GFP-FOXO3 reporter plasmid. Interestingly, UACC-62 parental cells tolerated the overexpression of FOXO3 better than the cells without TRIB2. FOXO3 was found to be localized both in the cytoplasm and the cell nucleus in the parental cells, while localization in the few surviving TRIB2 KO cells was less consistent and with more cells with nuclear fluorescence (Appendix A). Western blot analysis of parental and TRIB2-KO UACC-62 cell lines revealed that phosphorylation of AKT and FOXO was decreased in untreated UACC-62 cells without TRIB2 (Figure 3F and Appendix A). Unlike U2OS cells, UACC-62 cells did not show altered levels of total FOXO3 protein levels upon compound treatment. Exposure of the isogenic cell lines to BEZ235 almost completely abolished AKT phosphorylation, while HAR and PIP reduced it significantly after 2 h of treatment. Importantly, the effect of HAR and PIP treatment on AKT and FOXO phosphorylation was even more pronounced in TRIB2 KO cells compared to parental cells. These data are in agreement with the previously established role of TRIB2 as an activator of AKT [2] and as a FOXO repressor [8].

Next, we treated UACC-62 isogenic cell lines with BEZ235 (100 nM) for 72 h and analyzed the transcript level of FOXO target genes. BEZ235 treatment significantly increases the transcriptional levels of the FOXO targets PLK1, OCT4 and PUMA in cells lacking TRIB2 expression when compared to cells that express TRIB2 (Figure 3G and Appendix A). Additionally, we show that UACC-62 cells expressing high levels of TRIB2 induce p21 and PUMA transcription upon HAR and PIP treatment, respectively (Figure 3G). Furthermore, the absence of TRIB2 expression in TRIB2 KO cells potentiates the effect induced by HAR and PIP treatment (Figure 3G). These data together indicate that upregulation of the FOXO target genes p21 and PUMA by HAR, PIP and BEZ235 is affected by TRIB2 levels.

## 3. Discussion

TRIB2 confers resistance to several anti-cancer therapies and, therefore, pharmaceutical means to interfere with its activity might have the potential to overcome drug resistance and improve clinical outcome in cancer patients. Here, we used RNA sequencing-based transcriptional profiling of isogenic cells lines with different TRIB2 statuses in the presence or absence of PI3K/mTOR inhibition to identify sets of differentially expressed genes. Using cMAP algorithms, we showed that several small-molecule compounds induce inverse signatures, suggesting their potential to interfere with TRIB2 downstream effects. RNA-seq analysis showed that the transcriptional signature of the U2OS osteosarcoma cell line is altered by the overexpression of TRIB2 and exposure to the PI3K/mTOR inhibitor BEZ235. The inhibition of PI3K/AKT signaling by BEZ235 treatment affects the expression level of over 2000 genes, which is in line with previous studies [9], whereas the ectopic expression of TRIB2 affects ten times fewer genes. This might be because TRIB2 only affects certain aspects of downstream signaling pathways, e.g., phosphorylation at serine 473 in AKT. In general, TRIB2 overexpression increased the expression of genes that promote tumor progression, while it repressed those genes known to prevent tumor progression. Importantly, and in line with previous studies, PI3K/mTOR inhibition by BEZ235 treatment showed an opposite transcriptional signature [9,52]. TRIB2 expression level is associated with increased resistance to different chemotherapeutic drugs [2,13,14,15,16,17]. In this study, we used connectivity map algorithms to identify compounds showing inverse similarities between a TRIB2-induced expression signature and the reference profile aiming at reversing drug resistance. The potential of this approach has been clearly illustrated by numerous investigations performed to identify novel indications for approved drugs or compounds with known modes of action [53,54,55]. Accordingly, we found that the naturally occurring alkaloids HAR and PIP are capable of reverting TRIB2-dependent transcriptional signatures. From these results, we found that HAR and PIP act synergistically with the PI3K/mTOR inhibitor BEZ235, affecting cell viability. Thus, cells with high levels of TRIB2 treated with either of these compounds would be more prone to enter apoptosis or become more senescent upon treatment with PI3K inhibitors. Furthermore, we determined that both compounds act upstream of FOXO, regulating its subcellular localization and transcriptional activity. In agreement with our data, PIP has been reported to inhibit the PI3K/AKT axis in human triple-negative breast cancer cells [56] and to activate FOXO3, inducing the expression of the FOXO target gene BCL2 Like 11 (BIM) in HeLa cells [57]. Furthermore, PIP was reported to bind covalently to a cysteine residue within the export receptor protein CRM1 and inhibit the nuclear export of proteins that bear nuclear export sequences [58]. PIP is found in the fruit of long pepper (*Piper longum*) and has been used in traditional Indian medicine for the treatment of many diseases, including tumors [59]. In line with our data, PIP has been found to reverse chemotherapy resistance in several cancers [60,61,62,63]. Interestingly, a recent study showed PIP to reverse resistance to cisplatin in non-small cell lung cancer cells by inhibiting AKT phosphorylation [64].

HAR is a fluorescent β-carboline alkaloid found in several plant species, including *Banisteriopsis caapi* vine and *Peganum harmala*, and has been shown to act as a reversible, selective inhibitor of monoamine oxidase (MAO)-A [65], a flavoenzyme that degrades amine neurotransmitters by oxidative deamination. It remains to be determined whether the enzymatic inhibition of MAO-A mediates the reversion of the TRIB2-induced gene expression profile and the synergic susceptibility observed upon HAR treatment. Reversible inhibitors of monoamine oxidase A (RIMAs) chemically unrelated to HAR, such as moclobemide, brofaramine, toloxatone or befloxatone, could be used to test this hypothesis. Conversely, HAR has been shown to potently and specifically inhibit the kinase activity of DYRK1A in vitro and cultured cells [66]. Interestingly, DYRK1A kinase is known to phosphorylate FOXO1 at Serine 329 [67]. A more recent study shows that inhibition of DYRK1A decreased phosphorylation of FOXO3 on Ser253 by downregulation of AKT activity in head and neck squamous cell carcinoma cell lines [68]. Taken together, these data, together with our findings, suggest that both of the natural product alkaloids PIP and HAR might be very useful in the clinical setting to treat cancer and to overcome therapy resistance.

## 4. Materials and Methods

### 4.1. Cell Culture

The human osteosarcoma cell line U2OS and the human melanoma cell line UACC-62 were purchased from American Type Culture Collection (ATCC) and maintained in Dulbecco’s Modified Eagle Medium (DMEM) (Sigma, Faro, Portugal) supplemented with 10% Fetal bovine serum (FBS) (Sigma, Faro, Portugal) and antibiotics (Invitrogen, Carlsbad, CA, USA). Cell cultures were maintained in a humified incubator at 37 °C with 5% CO_2_ and passaged when confluent using trypsin/EDTA. U2OS-TRIB2 and U2OS-empty cell lines were generated by stably transfecting pEGFP-N1-TRIB2 and pEGFP-N1, respectively, into parental U2OS cells. Stable cell lines U2nesRELOC and U2foxRELOC cells were generated as described previously [46,47,69].

### 4.2. hTRIB2 KO Cell Generation

We used clustered regularly interspaced short palindromic repeats (CRISPR)/CRISPR associated protein 9 (Cas9) to genetically disrupt the TRIB2 locus in the UACC-62 melanoma cell line. The two guide RNAs (gRNAs), targeting exon 1, were cloned into a plasmid containing Cas9 and pSpCas9(BB)-2A-Puro (PX459) V2.0. Plasmids were transfected into UACC-62 cell line with Lipofectamine 2000 (Thermo Fisher Scientific, Faro, Portugal) and cells were selected following 48 h treatment with 1 µg/µL puromycin (VWR, Faro, Portugal). Single-cell clones were expanded and analyzed by Western blot.

### 4.3. Cell Transfection and Selection

The parental UACC-62 and TRIB2 KO cells cultured as described above but without antibiotics were transfected using Lipofectamine 2000 (Thermo Fisher Scientific, Faro, Portugal) with pEGFP-C3-FOXO3 and selected with 1 μg/mL of neomycin. Clones were grown and GFP-positive clones were selected.

### 4.4. RNA Sequencing

Total RNA was extracted by using TRI-reagent (Sigma, Faro, Portugal). Library preparation and sequencing of RNA was performed at the Centro Nacional de Investigaciones Cardiovasculares (CNIC) Genomics Unit (Madrid, Spain). All samples were prepared in biological triplicate. RNA was sequenced with the GAIIx sequencer. The sequencing protocol was single-end 75-bp elongation.

### 4.5. Mapping and Determination of DEGs and Gene Enrichment Analysis

Raw data from the RNA sequencing consisted of the four conditions in triplicate. Sequencing adaptor contaminations were removed from reads using Cutadapt software [70] and the resulting reads were mapped and quantified on the transcriptome (Ensembl GRCh37.v72) using RSEM v1.2.3 [71]. Only genes with at least 1 count per million in at least 3 samples were considered for statistical analysis. Data were then normalized, and differential expression was tested for the comparisons depicted in Figure 1A using the Bioconductor package EdgeR [72]. We considered those genes with a Benjamini–Hochberg adjusted *p*-value ≤ 0.05 as differentially expressed. 

Enrichr computes three types of enrichment scores to assess the significance of the overlap between the input list and the gene sets in each gene set library. These tests are (1) the Fisher exact test, a test that is implemented in most gene list enrichment analyses; (2) the z-score, a correction test that computes the deviation from the expected rank by the Fisher exact test; and (3) a combined score that multiplies the log of the *p*-value computed with the Fisher exact test by the z-score computed by our correction to the test. We selected the combined score to correct for Fisher’s exact test [35,36].

cMAP analysis was performed by querying experiments with drugs that generated transcriptional signatures similar and reversed to the TRIB2 transcriptional signature. Drugs were selected with *p*-value < 0.05 and lower enrichment scores. The enrichment scores were calculated based on the Kolmogorov–Smirnov statistic, a non-parametric rank statistic, also known as gene set enrichment analysis (GSEA), to interpret gene expression data [38,39]

### 4.6. Quantitative Real-Time PCR

To determine the transcription levels of genes affected by TRIB2, BEZ235, HAR and PIP, cells were treated with each drug alone, 24 h after plating, at a cell confluency of 70–90% on 10-cm plates (20101, SPL, Faro, Portugal). After the treatment duration, cells were collected to extract RNA the with E.Z.N.A. Total RNA Kit I (R6834-02, VWR, Faro, Portugal), according to the supplier’s instructions. Synthesis of cDNA was performed with the NZY First-Strand cDNA Synthesis Kit (MB12502, Nzytech, Faro, Portugal) according to the supplier’s instructions. Quantitative PCR was performed on a CFX96 Real-Time System (C1000™ Thermal Cycler, Biorad, Faro, Portugal) using Applied Biosystems’ SYBR™ Green PCR Master Mix (4309155, Thermo Fisher Scientific, Faro, Portugal). Relative quantification of gene expression was determined by the 2^−ΔΔCt^ method [73].

### 4.7. MTT Assay

Cell viability was determined using the MTT colorimetric assay. Four thousand cells were plated in a 96-well plate and treated the following day with the respective drugs for 48 and 72 h. Cell media were removed and replaced with 100 uL of fresh media containing MTT (0793, VWR, Portugal) solution at a final concentration of 0.5 mg/mL. After 3 h of incubation, the media were aspirated and replaced with 100 uL of DMSO to dissolve the formazan crystals. Absorbances were measured at 560 nm (Abs560) and the reference wavelength 700 nm (Abs700). The “No cells” control consisted in MTT being added to wells without cells. Absorbances at 560 nm were normalized by subtracting the reference wavelength values (Norm Abs560 = Abs560 sample − Abs700 sample). Cell viability % = (Norm Abs560 sample − Norm Abs560 blank)/(Norm Abs560 control − Norm Abs560 blank) × 100. 

### 4.8. Drug Synergy Assay

We determined dose–response curves for BEZ235 (Dactolisib) (Selleckchem, Houston, TX, USA), HAR (ACRO302972500, VWR, Faro, Portugal) and PIP (CAYM11006-25, VWR, Faro, Portugal) in the U2OS-TRIB2 cells to generate equipotent concentration ratios of BEZ235/HAR and BEZ235/PIP. Twenty-four hours after plating, U2OS-TRIB2 cells were treated with 20 serial dilutions of the compounds for 48 and 72 h. We determined cell viability at each time point with MTT assays. We generated normalized dose–response plots (log (drug [µM] vs. cell viability (%)) and non-linear regression to calculate GI50 values. We evaluated the synergistic effect on cell viability of BEZ235 and HAR co-treatment by treating U2OS-TRIB2 cells with five serial dilutions of each drug alone and both drugs combined, at a constant ratio by 2-fold increase. The same protocol was performed for BEZ235 and PIP synergy experiments. Cell viability was determined using MTT assay. Synergy values were obtained using the CompuSyn software [74], based on the Chou Talalay method. Combination index (CI) < 1 denotes synergism, CI > 1 antagonism and CI = 1 additive.

### 4.9. FOXO Translocation Assay

The U2foxRELOC system is a FOXO translocation assay that has been previously established [47,75]. Briefly, cells were seeded at a density of 40,000 cells per well onto coverslips covering a 24-well plate (SPL Life Sciences Co, Gyeonggi, Korea). The following day, cells were treated with HAR or PIP for 1 h. Cells were fixed with 4% paraformaldehyde for 10 min and washed 3 times with 1 × phosphate-buffered saline (PBS). The next day, the coverslips were mounted on slides using mounting media coupled with DAPI (Santa Cruz Biotechnology, Santa Cruz, CA, USA) to stain the nucleus and stored at 4 °C. All experiments were performed in triplicate. Samples were imaged using 40× lenses on the AxioImager Z2 microscope (Zeiss) imaging system. Images were obtained using AxioVision 4.8.2 software.

### 4.10. Image and Data Analysis

Quantification of FOXO localization was performed by Definiens Developer v2.5 software (Definiens, Munich, Germany), Nucleus and cytoplasm segmentation was performed with a custom-made ruleset using a nuclear DAPI signal for the nucleus and then growing the area to identify the cytoplasm. After the segmentation, the ratio of green intensity was measured in both the nucleus and cytoplasm, and then the ratio of nucleus to cytoplasm was calculated to define a threshold for translocation.

### 4.11. Western Blot

For the preparation of whole-cell lysate, cells were harvested and lysed in a lysis buffer (20 mM Tris pH 7.5, 150 mM NaCl, 1% Triton X-100, 50 mM NaF, 1 mM EDTA, 1 mM ethylene glycol-bis(β-aminoethyl ether)-*N*,*N*,*N*′,*N*′-tetraacetic acid (EGTA), 2.5 mM sodium pyrophosphate, 1 mM b-glycerophosphate, 10 nM Calyculan A and EDTA-free complete protease inhibitor cocktail (PIC) (Sigma, Faro, Portugal). A sample buffer was added to 1X final, and samples were boiled at 95 ℃ for 5 min. Samples were resolved on 8–12% SDS-PAGE gels, transferred to nitrocellulose membranes and immunoblotted according to the antibody manufacturer’s instructions. Secondary antibodies were added (GE Healthcare) at typically a 1:10,000 dilution for 1 h at room temperature. Visualization of signals was achieved using the ChemiDocXRS þ Imaging System (BioRad). Anti-FOXO3a (#2497), P-AKT Ser 473 (#4060) and total AKT (#9272) were purchased from Cell Signaling Technology (CST, Faro, Portugal) and P-FOXO Ser 253 (sc-101683) and GAPDH FL-335 (sc-25778) were purchased from Santa Cruz Biotechnology (SCB, Faro, Portugal). 

### 4.12. Trypan Blue Exclusion Assay

Cells were resuspended and mixed with trypan blue (1:1 *v*/*v*). Viable (unstained) and unviable (stained) cells were counted on a hemocytometer and multiplied by two (dilution factor). Cell viability percentage was calculated by dividing the total number of live cells/mL by the total number of cells/mL and multiplied by 100 [76].

## 5. Conclusions

TRIB2 confers resistance to several anti-cancer drugs, including the PI3K/mTOR inhibitor BEZ235, by inactivating the AKT/FOXO signaling axis. We show, here, by using RNA sequencing, that transcriptional signatures are altered by the presence of TRIB2 or the exposure to the PI3K/mTOR inhibitor BEZ235, and we identified the small-molecule compounds HAR and PIP, capable of reversing these signatures. Furthermore, these compounds induce the nuclear translocation of FOXO3 and synergized with BEZ2335, decreasing TRIB2-mediated resistance to BEZ235 treatment.

## Figures and Tables

**Figure 1 cancers-12-03689-f001:**
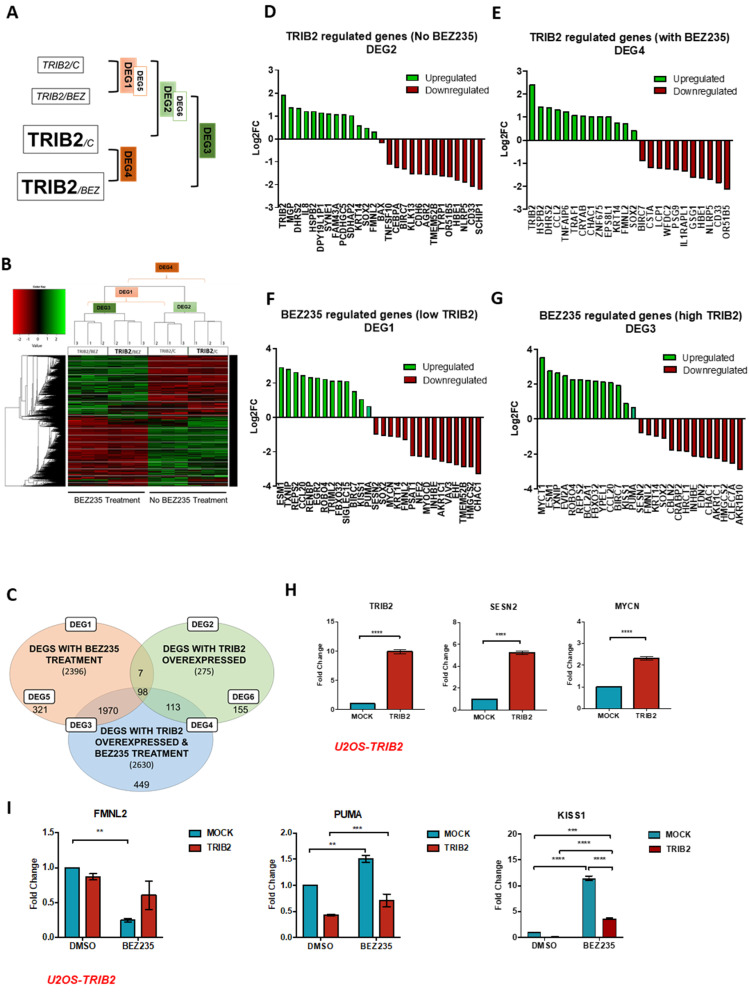
Tribbles homologue 2 (TRIB2) overexpression induces transcriptional changes. (**A**) Schematic representation of the experimental conditions and the differentially expressed genes (DEGs) analyzed. DEG1 refers to genes differentially expressed by BEZ235 treatment in cells without TRIB2, and DEG2 refers to genes differentially expressed by TRIB2 in untreated cells. DEG3 refers to genes differentially expressed by BEZ235 treatment, regardless of TRIB2 levels, while DEG4 represents genes that are differentially expressed by TRIB2, with or without BEZ235 treatment. DEG5 consists of genes exclusively regulated by BEZ235 treatment in cells with low TRIB2 expression and is, thus, a subset of DEG1, while DEG6 refers to genes exclusively regulated by TRIB2 in untreated cells and is, hence, a subset of DEG2. RNA-seq was performed with isogenic U2OS cell lines that stably express either pEGFP-N1 or pEGFP-N1-TRIB2. Cells were treated with 100 nM BEZ235 or vehicle (DMSO) for 72 h. (**B**) Heat map displaying the transcriptional signatures of BEZ235 treatment and TRIB2 overexpression. Upregulated genes (green) and downregulated genes (red) are shown for U2OS cells, untreated and BEZ235-treated, with low and high TRIB2 levels. DEG2 and DEG3 result from the comparison of the transcriptional signatures of U2OS cells with low and high TRIB2 levels in untreated and BEZ235-treated cells, respectively, while DEG1 and DEG4 result from the comparison between untreated and BEZ235-treated cells with low and high TRIB2 levels, respectively. Genes were considered differentially expressed with a Benjamini–Hochberg adjusted *p*-value ≤ 0.05. (**C**) Venn diagram showing the number of DEGs by TRIB2 overexpression (green balloon, DEG2), by BEZ235 treatment (pink balloon, DEG1) and by both (blue balloon). The number of genes in overlapped areas consist of common genes that are affected by combinations of two or three conditions. The lists of genes were obtained assuming a cut-off of log2(fold change) = 0.5 on all significantly differentially expressed genes (adjusted *p*-value ≤ 0.05). (**D**–**G**) Top differentially expressed genes by TRIB2 in untreated cells (**D**) and BEZ235-treated cells (**E**) and by BEZ235 treatment in cells with low TRIB2 (**F**) and high TRIB2 (**G**–**I**). Validation by RT-qPCR of top DEGs from the RNA-seq. Results represent the mean ± SEM generated from three independent experiments with triplicates. (**H**) mRNA levels were detected in U2OS cells with low TRIB2 (MOCK) and with TRIB2 overexpressed. TRIB2, SESN2 and MYCN are upregulated in high TRIB2 U2OS cells. Statistical significance was determined by unpaired *t*-test with two-tailed *p*-value < 0.05, with (****) *p* < 0.0001. (**I**) cells were treated with vehicle (DMSO) or BEZ235 100 nM for 72 h. BEZ235 downregulated FMNL2 and upregulated PUMA and KISS1 on MOCK cells. The presence of TRIB2 decreased BEZ235 modulation of FMNL2, PUMA and KISS1. Statistical significance was determined by two-way ANOVA with Tukey’s multiple comparisons test, with (**) *p* < 0.01, (***) *p* < 0.001, (****) *p* < 0.0001.

**Figure 2 cancers-12-03689-f002:**
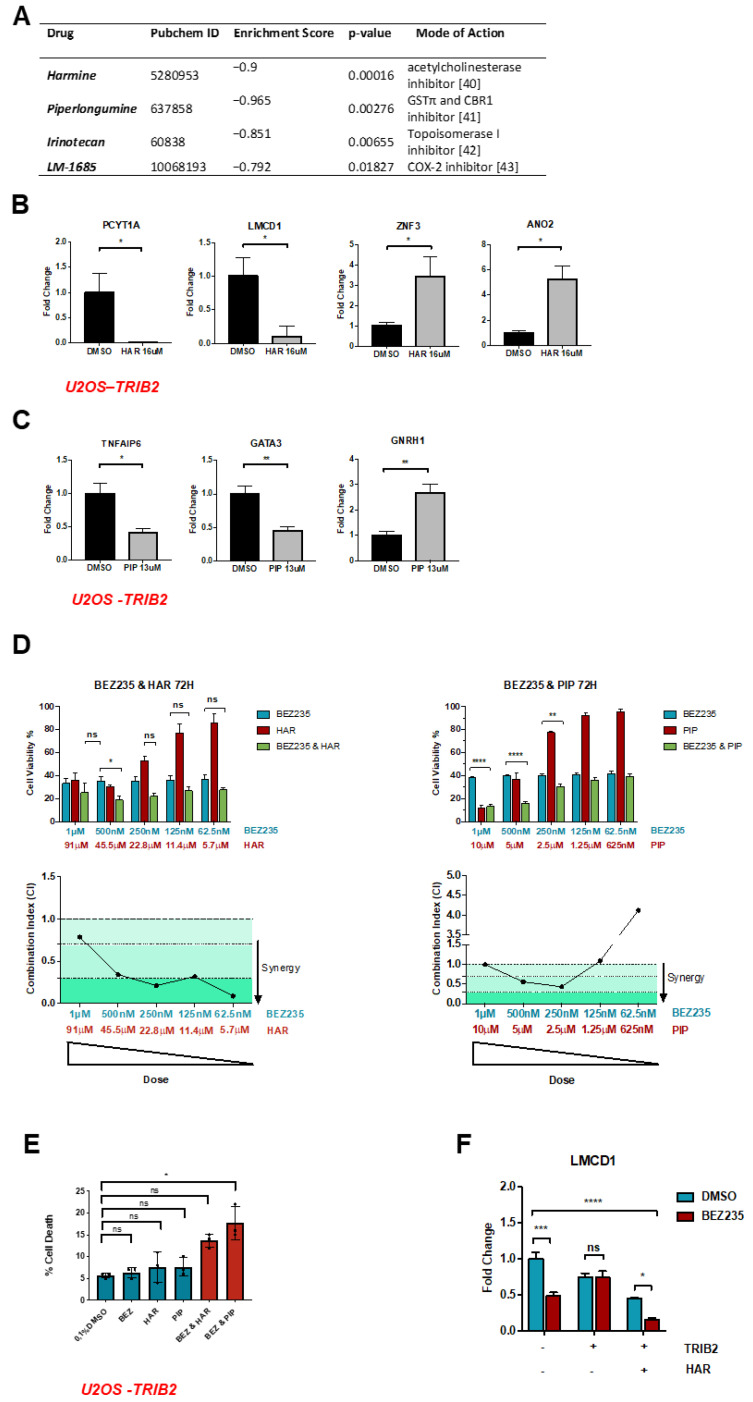
Harmine (HAR) and piperlongumine (PIP) reverse TRIB2-induced expression profiles. (**A**) A connectivity map (cMAP) query was performed to identify drugs that generated reversed transcriptional signatures of TRIB2 on MCF7, PC3 and MDA-MB-231 cell lines. Drugs were selected according to *p*-value < 0.05 and lower enrichment score. (**B**) HAR’s top regulated genes from cMAP analysis were validated by RT-qPCR in U2OS-TRIB2 cells. HAR upregulates ANO2 and ZNF3 and downregulates LMCD1. (**C**) PIP’s top regulated genes from cMAP analysis were validated by RT-qPCR. PIP upregulates GNRH1 and downregulates TNFAIP6 and GATA3. (**B**,**C**) results represent mean ± SEM from three independent experiments in triplicate. Statistical significance was determined by an unpaired *t*-test with one-tailed *p*-value, with (*) *p* < 0.05, (**) *p* < 0.01 and (ns) *p* > 0.05. (**D**) U2OS-TRIB2 cells were treated with BEZ235, HAR or PIP alone, BEZ235 and HAR or BEZ235 and PIP combined for 72 h. Cell viability was determined by measuring cellular metabolic activity using a MTT assay. BEZ235 and HAR displayed synergy in all concentrations, while BEZ235 and PIP displayed synergy at combined concentrations of 250 nM (BEZ235)/2.5 µM (PIP) or higher. Synergy was determined by the Chou Talalay method, where combination index (CI) < 1 denotes synergism, CI > 1 antagonism and CI = 1 additive. Statistical significance was determined by a two-way ANOVA with Tukey’s multiple comparisons test, with *p*-value < 0.05, (ns) *p* > 0.05, (*) *p* < 0.05, (**) *p* < 0.01, (****) *p* < 0.0001. (**E**) A cell-death assay was performed on U2OS-TRIB2 cells treated with 250 nM BEZ235 and 22.8 µM HAR (the chosen synergic concentration) or 250 nM BEZ235 and 2.5 µM PIP (the chosen synergic concentration) for 72 h. The cell-death assay displays an increase in cell death when cells are treated with combined drugs, compared to the DMSO control, in line with the MTT results. *p*-values were obtained from an unpaired *t*-test with Welch’s correction, (*) *p* < 0.1, (ns) *p* > 0.05. The mean ± SEM from three independent experiments is shown. (**F**) RT-qPCR analysis of LMCD1 gene in U2OS-TRIB2 cells treated with 250 nM BEZ235 and 22.8 µM HAR (the chosen synergic concentration) or 250 nM BEZ235 and 2.5 µM PIP (the chosen synergic concentration) for 72 h. Statistical significance was determined by a two-way ANOVA and Tukey’s multiple comparisons test to evaluate the contribution of TRIB2 and by a two-way ANOVA and Sidak’s multiple comparisons test to evaluate the contribution of BEZ235 to gene expression, with (*) *p* < 0.05, (***) *p* < 0.001, (****) *p* < 0.0001, (ns) *p* > 0.05.

**Figure 3 cancers-12-03689-f003:**
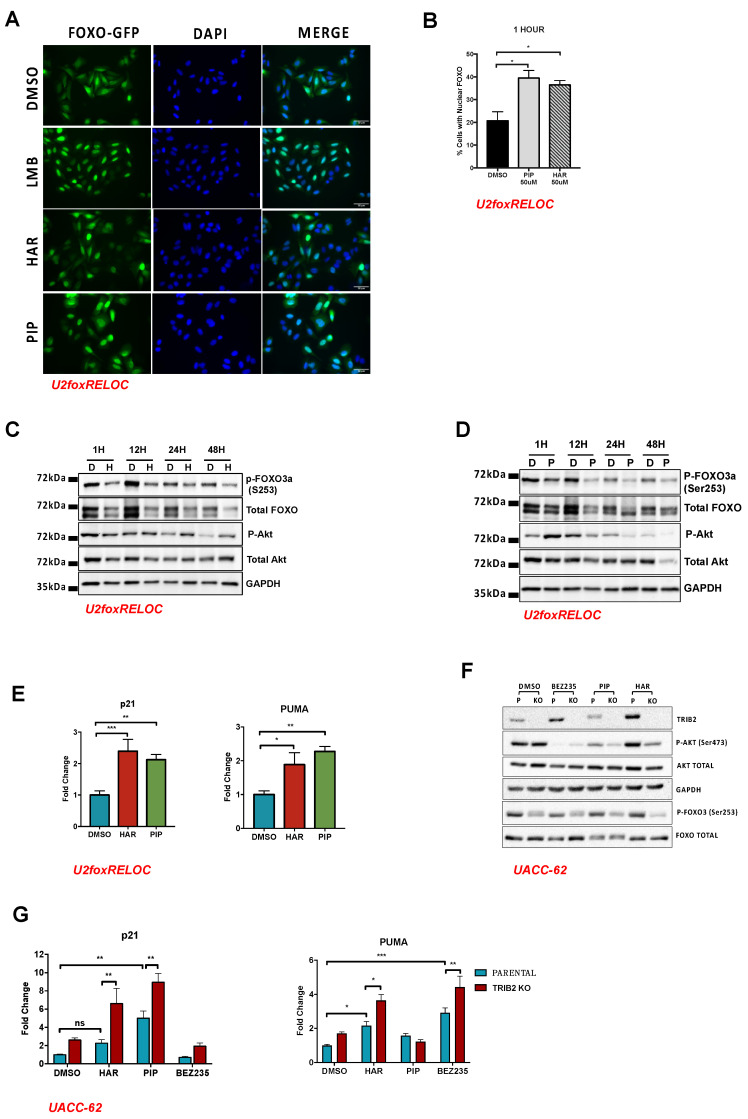
HAR and PIP induce FOXO nuclear translocation. (**A**) U2foxRELOC cells were treated with 50 µM HAR or PIP for 1 h. DMSO and Leptomycin B (LMB) were used as negative and positive controls, respectively, for FOXO localization. DMSO-treated cells continued to display FOXO mainly in the cytoplasm, while LMB-, HAR- and PIP-treated cells displayed FOXO mainly in the nucleus. (**B**) Quantification of FOXO localization was performed by determination of the ratio of green intensity measured in both nucleus and cytoplasm. Cells treated with HAR or PIP for 1 h showed a two-fold increase in nuclear FOXO compared to DMSO-treated cells. Statistical significance was determined by a one-way ANOVA with Dunnet’s multiple comparisons test, and (*) *p*-value < 0.05. (**C**,**D**) The U2foxRELOC cell line was treated with 50 µM HAR (H) in (**C**) or 50 µM PIP (P) in (**D**) for 1, 12, 24 and 48 h and Western blot analysis was performed to evaluate the levels of FOXO3a and P-FOXO3a and Akt and P-Akt. DMSO (**D**) was used as a control for HAR and PIP treatments. Glyceraldehyde-3-Phosphate Dehydrogenase (GAPDH) was used as loading control. Representative image of two independent experiments. (**E**) U2foxRELOC cells were treated with HAR 16 µM, PIP 13 µM or DMSO (control) for 6 h. PUMA and p21 mRNA levels were determined by RT-qPCR. Graphics represent the mean ± SEM from four independent experiments in triplicate. Statistical significance was determined by a one-way ANOVA with Tukey’s multiple comparisons test and with (*) *p* < 0.05, (**) *p* < 0.01, (***) *p* < 0.001 and (ns) *p* > 0.05. (**F**) Parental UACC62 cells (P) and UACC-62 TRIB2 knockout cells (KO) were treated with 16 µM HAR or 13 µM PIP for 2 h. Proteins from lysates were immunoblotted with phospho-specific antibodies, as indicated. (**G**) UACC-62 parental and UACC-62 TRIB2 KO cells were treated with HAR 16 µM, PIP 13 µM and BEZ235 100 nM for 12 h. The relative expression of PUMA and p21 was assessed by RT-qPCR and is expressed as fold change relative to DMSO treatment. *p*-values were obtained from a two-way ANOVA with Sidak’s and Dunnet’s multiple comparisons tests; (*) *p* < 0.05, (**) *p* < 0.01 and (***) *p* < 0.001, (ns) *p* > 0.05. The mean ± SEM from three independent experiments in triplicate is shown.

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
