# Peer review of "Harmine and Piperlongumine Revert TRIB2-Mediated Drug Resistance"

_cancers, 2020, doi:10.3390/cancers12123689_

Round 1
Reviewer 1 Report
I have checked the rebuttal letter and all comments are satisfied by the authors. Thus, this manuscript is now warrant for publication by your esteemed journal.
Reviewer 2 Report
to me, the authors have addressed in a very complete way all the comments made by reviewer 1. They brought new experiments and data. So, based on reviewer 1's comments and author's reply, the paper looks to me suitable for publication.
This manuscript is a resubmission of an earlier submission. The following is a list of the peer review reports and author responses from that submission.
Round 1
Reviewer 1 Report
Machado et al show very interesting work on TRIB2 mediated chemoresistance to PI3K/mTOR inhibitor BEZ and provide a drug combination that can potentially revert this chemoresistance. This is a good study however there are a few issues with the data as presented, which do not support the conclusions. The RNA-seq data and the drug screen is very interesting, but the concluded mechanistic link to FOXO, AKT and FOXO target genes is not supported by the experiments shown.
- What concentration of BEZ was used, and what time point used for the RNA-seq in figure 1, and how long are the isogenic cells expression TRIB2, and have they been selected (100% of cells expressing TRIB2)? The legend should contain this information.
- Figure 1: This analysis is done in U20S cells, so what is the basal level of TRIB2 expression in these cells – is it zero or is there already some expression in these cells, and is the level of expression in the control and TRIB2 overexpressed cells at a physiological level comparable to that found in normal cells or other tumour cells for example?
- For figure 1, when comparing the DEGs from the comparisons, is there a list of DEGS that are unique to the TRIB2 overexpression control and treated arm? i.e. are there genes that are changed only when TRIB2 is overexpressed? (DEG4 list minus DEG1 list – was this list created). It would more interesting to compare and discuss in the text (extending from statement in line 150) the DEGs in the context of what is changed when cells are treated with BEZ (sensitive) and what changes occur when TRIB2 is overexpressed with treatment (now resistant), to really pinpoint DEGs that may be up or down and correlate with the chemoresistant phenotype you have previously reported (ref 3).
- It really isn’t clear which cells are referred to as “low TRIB2” – are these the control mock transduced cells? Why not refer to cells as NT (controls, not transduced) rather than low TRIB2. For figure 1D-G, these figures should also indicate which DEG1-4 they are from, in addition to the plot titles to avoid confusion.
- Figure 1D-G graphs are the top differentially DEGs -what criteria defines the “TOP” genes and the number of genes which were included in the graphs? It is very good validation that CEBP expression levels are down in figure 1D, consistent with the literature, which could be pointed out in the text.
- Regarding the expression level query in question 2, what effect does knockdown have in U2OS cells on the genes selected in figure 1? This may not have been possible and hence you used the CRISPR approach in another line – so same question then for the gene expression levels for those selected in figure 1 in these cells?
- If I understood correctly, the DEG list used to screen the CMAP was DEG2 (explained in line 159), and the purpose is to find a drug that does the inverse as TRIB2 overexpression in the absence of BEZ, in the hope that a drug is found that would syngergise with BEZ and revert TRIB2 mediated chemoresistance? Can this be more clearly stated in line 159.
- Line 162: The list of candidate drugs was obtained by ruling out enrichment scores with P-values greater than 0.05. Can the enrichment score number be defined/explained please.
- Line 203 refers to figure 2D which is the MTT assay, not gene expression as referred to in the text. Please correct and refer to figure 2 D and supplemental figure 2D correctly (both the same data only different time points). Also, can it be clearly distinguished when referring to the MTT assay used to generate the CIs, that this is a metabolic assay not a viability assay. Are the chosen synergistic concentrations used in figure 2E those from the MTT at which time point? Line 214 refers to viability also for figure 2E: figure 2e is gene expression. Cell viability by a flow assay or cell count using trypan blue exclusion at the very least should be shown in figure 2 to show that there is synergistic cell kill. This section incorrectly states that viability has been assessed for synergy when it has not. Figure 2F is discussed in the text but doesn’t exist, which is actually shown as figure 2E – so this reviewer is wondering if a cell viability figure is missing?
- Define what you mean by GI50 in line 208?I assume a cytostatic effect given that the MTT assay was used. I think it should be determined that these drugs kill, and the EC50 determined – therefore confirming a cytotoxic effect.
- Line 274 concludes that FOXO phosphorylation is impaired by the 2 drugs. It is not clear from the data however. It appears total FOXO is lost, negating any effect by Har in figure 3C. And only the 12 and 24 h time point agree with this statement in figure 3D with PIP. The authors would need to refine the time point and data shown for PIP to support this statement, and reassess their conclusion for figure 3D with HAR.
- P21 and PUMA can be regulated by a number of factors, not just FOXO and the authors have not clearly shown that the genes are changed by HAR and PIP due to FOXO regulation. Their conclusion line 290 is not fully supported.
- The inclusion of the isogenic TRIB2 KO line in figure 3F is a good addition – however, the effect of cell viability and cell kill in these lines, with the drugs at the concentrations used has not been shown. It is difficult to use the same lines for overexpression and knockout depending on the level of gene expression – however each cell line will have its own inherent drug resistance depending on its genetic background – so it is difficult to match the knockdown data with the overexpression data and mechanistically linking this with FOXO and TRIB2. To support this the authors need further experiments including at least:
- Figure 3 A extended to show FOXO nuclear translocation with HAR and PIP in cells with and without TRIB2 overexpression (or knocked out)
- Cell death assay showing the cytotoxic synergy of BEZ with HAR or PIP
- FOXO and AKT activity (by phosphorylation western blotting) in the presence of HAR or PIP +/- BEZ or +/-TRIB2 overexpression.
Reviewer 2 Report
By using RNA sequencing-based transcriptional profiling of isogenic cells lines and pattern-matching algorithms of Connectivity Map, Machado et al identified several genes involved TRIB2 regulatory pathway and screened two compounds Harmine and Piperlongumine that could invert the TRIB2-associated expression profile. These findings are of interesting. However, there are some issues underlying the experiments performed here that need to be further strengthened for supporting their conclusion. Below I list my concerns, that:
- Why do authors use U2OS, osteosarcoma, as model cells? The rationale should be introduced in detail.
- Despite data showed that HAR and PIP may help cells regain sensitivity to BEZ235 toxicity in the presence of high levels of TRIB2. Two important issues should be considered here. First, Is BEZ235 used in treating osteosarcoma in clinic? At least, Cisplatin but not BEZ235 is often selected as chemotherapeutic drug in treating osteosarcoma. Thus, I cannot understand why authors consider BEZ235 rather than cisplatin in studying drug resistance of osteosarcoma. Second, high level of TRIB2 dose not mean these cancer cells are resistant to BEZ235. Thus, according to the title of this manuscript “Harmine and Piperlongumine revert TRIB2-mediated drug resistance”, authors should establish TRIB2-mediated resistant cell line, or alternatively, provides data to support TRIB2 overexpressing U2OS cells are resistant to BEZ235.
- As FOXO nuclear translocation is accompanied by loss of phosphorylation on Ser253 by AKT, authors monitored the phosphorylation status of FOXO and AKT upon HAR and PIP treatment and concluded that HAR and PIP treatment impairs FOXO phosphorylation. This effect seems to be regulated by AKT activation status. Upon the treatment of HAR, the phosphorylation status of FOXO3a was not affected (Figure 3C); on the other hand, phosphorylated AKT increased. Besides, treating cells with HAR reduced total amount of FOXO3a, thus, the role of FOXO3a might independent of AKT phosphorylation. This phenotype was also observed in PIP-treated cells. Thus, in any event, direct evidence should be provided to show that FOXO3 translocation and AKT activation are interrelated.
- The immunoblots of supplementary figure 3B are not convincing.
Minor concerns
- In figure 3A, what are HARMINE and PIPER? The same, what are D and H in (C) and P in (D)? These should be explained in the figure legends.
- Please show the Standard Deviation of quantitative results in figure 3C and D.